# Genome-Wide Association Study of Lactation Traits in Chinese Holstein Cows in Southern China

**DOI:** 10.3390/ani13152545

**Published:** 2023-08-07

**Authors:** Minqiang Su, Xiaojue Lin, Zupeng Xiao, Yuanhang She, Ming Deng, Guangbin Liu, Baoli Sun, Yongqing Guo, Dewu Liu, Yaokun Li

**Affiliations:** 1College of Animal Science, South China Agricultural University, Guangzhou 510642, China; smq2054113415@163.com (M.S.); jasonlin1996@163.com (X.L.); xiaozupeng@stu.scau.edu.cn (Z.X.); syh15521170780@163.com (Y.S.); dengming@scau.edu.cn (M.D.); gbliu@scau.edu.cn (G.L.); baolisun@scau.edu.cn (B.S.); yongqing@scau.edu.cn (Y.G.); 2National Local Joint Engineering Research Center of Livestock and Poultry, South China Agricultural University, Guangzhou 510642, China; 3Guangdong Key Laboratory of Agricultural Animal Genomics and Molecular Breeding, South China Agricultural University, Guangzhou 510640, China

**Keywords:** genome-wide association study, Chinese Holstein cows, high temperature and humidity, lactation traits

## Abstract

**Simple Summary:**

Lactation traits are economically important traits in dairy production, with milk yield, milk fat percentage, and milk protein percentage being the main indicators of cow performance and milk quality. The interaction between environment and genetics has a great influence on the performance of dairy cows. Cows are susceptible to heat stress under conditions of high temperature and humidity, which in turn affects lactation traits. Therefore, the aim of this study was to identify single-nucleotide polymorphism loci and candidate genes related to lactation traits in Holstein dairy cows under conditions of high temperature and high humidity in southern China. Our study identified 23 single-nucleotide polymorphic sites significantly associated with lactation traits, and 10 key candidate genes associated with lactation traits. The results of this study may provide a theoretical basis for the molecular marker breeding of Chinese Holstein cows in high-temperature and high-humidity environments in southern China.

**Abstract:**

Lactation traits are economically important for dairy cows. Southern China has a high-temperature and high-humidity climate, and environmental and genetic interactions greatly impact dairy cattle performance. The aim of this study was to identify novel single-nucleotide polymorphism sites and novel candidate genes associated with lactation traits in Chinese Holstein cows under high-temperature and humidity conditions in southern China. A genome-wide association study was performed for the lactation traits of 392 Chinese Holstein cows, using GGP Bovine 100 K SNP gene chips. Some 23 single nucleotide polymorphic loci significantly associated with lactation traits were screened. Among them, 16 were associated with milk fat rate, 7 with milk protein rate, and 3 with heat stress. A quantitative trait locus that significantly affects milk fat percentage in Chinese Holstein cows was identified within a window of approximately 0.5 Mb in the region of 0.4–0.9 Mb on *Bos taurus* autosome 14. According to Gene Ontology and Kyoto Encyclopedia of Genes and Genomes analyses, ten genes (*DGAT1*, *IDH2*, *CYP11B1*, *GFUS*, *CYC1*, *GPT*, *PYCR3*, *OPLAH*, *ALDH1A3*, and *NAPRT*) associated with lactation fat percentage, milk yield, antioxidant activity, stress resistance, and inflammation and immune response were identified as key candidates for lactation traits. The results of this study will help in the development of an effective selection and breeding program for Chinese Holstein cows in high-temperature and humidity regions.

## 1. Introduction

Consumer demand for high-quality dairy products is increasing with improvements in the living standards of the population. Lactation traits are very important economic traits in dairy production [1]. Lactation traits such as milk yield (MY), milk fat percentage (FP), and milk protein percentage (PP) are the main indicators of cow productivity and milk quality [2,3]. Changes in the environment, especially high temperature and humidity, have a significant impact on cow production performance [4]. Southern China has a subtropical climate with long periods of high temperatures and high humidity, and cows are subjected to heat stress in such environments. The effects of heat stress on dairy production can be categorized into two different causes: effects caused by reduced voluntary feed intake due to heat stress, and direct physiological and metabolic effects of heat stress [5]. Heat stress reduces MY and alters FP and PP through both thus [6,7]. Heat stress has been shown to affect the expression of genes related to milk fat metabolism [8]. Genomic regions associated with heat tolerance in dairy cows (Bos taurus autosome (BTA)5, BTA14 and BTA15) were identified by Sigdel et al. [9].

Genome-wide association study (GWAS) plays an important role in dairy breeding by helping to identify key genes, speeding up the breeding process and improving the genetic improvement of target traits, and many candidate genes and variant sites associated with lactation traits have been identified in Chinese Holstein cows by using GWAS. GWAS analysis of five milk yield traits in a Chinese Holstein cow population was performed by Jiang et al. [10]. A total of 105 significant single nucleotide polymorphism (SNP) loci were screened and localized to *DGAT1*, *GHR*, *CYP11B1*, *ABCG2*, and *LGB*. A GWAS was performed by Iung et al. [11] using 56,256 SNPs from 1067 cows to screen eight genes significantly associated with lactation traits. However, few GWAS have been reported for Chinese Holstein cows under high temperature and humidity conditions. Therefore, the aim of this study was to identify SNP sites and candidate genes related to lactation traits in Chinese Holstein dairy cows under high temperature and humidity conditions and provide a theoretical basis for molecular marker breeding of Chinese Holstein cows in high temperature and humidity environments.

## 2. Materials and Methods

### 2.1. Animal and Phenotypic Data

The Chinese Holstein cows used in the study were obtained from a dairy farm in Hezhou, Guangxi, China. The local climate is subtropical monsoon, with average annual temperatures ranging from 19.2 to 19.9 °C and average annual precipitation of 1558.1 to 2012.1 mm. All animals were raised under the same conditions, and healthy cows without diseases are screened according to the records of cattle diseases of pasture veterinarians. The experimental population consisted of 392 Chinese Holstein cows, and the pedigree of the cows can be traced back at least three generations. MY data were farm milk hall statistics. (For cows in the current lactation period with a lactation duration exceeding 305 days, the 305-day milk yield was determined based on actual statistical data. Conversely, for cows with a lactation duration less than 305 days, the 305-day milk yield was estimated using lactation curve analysis.) FP and PP were obtained by averaging five measurements after milk samples were collected by the farm each month and sent to Dairy Herd Improvement Center (DHI, China) and analyzed in the laboratory.

### 2.2. Genotyping and Quality Control

Blood samples were collected from the experimental cows, and DNA was extracted from the blood samples by using a Magnetic Universal Genomic DNA Kit (TIANGEN, Beijing, China). Genotyping was performed using the GGP Bovine 100 K SNP GeneChip on the Illumina iScan SNP microarray typing platform. Quality control was performed using the PLINK software (v1.9) [12], and the quality control criteria were (1) exclusion of SNPs at unknown locations and on sex chromosomes, (2) exclusion of individuals with SNP deletions greater than 5%, (3) exclusion of SNPs with detection rates less than 95%, (4) exclusion of SNPs with sub-allele frequencies less than 0.05, and (5) exclusion of SNPs that deviated from Hardy–Weinberg equilibrium (*p* < 1.0 × 10^−6^).

### 2.3. Phenotypic Data Processing

Phenotypic data for lactation traits were collated. Abnormal values that deviated from the phenotypic mean ± 3 times the standard deviation were excluded. Fixed-effect levels were classified as follows: calving age was divided into six levels, i.e., <24, 25–36, 37–48, 49–60, 61–72, and >72 months; parity was divided into five levels, from the first to the fifth parity; calving season was divided into four levels, November–February, March–April, September–October, and May–August according to the average maximum temperature of January–December in Guangxi Hezhou City.

### 2.4. Fixed-Effects ANOVA

A general linear model with the ANOVA function of R 4.1 software was used to perform ANOVA of the fixed effects. Three effects of calving month, calving season, and litter size were considered for lactation traits, and the model was as follows:$$Y_{ijkl}=\mu +Month_{i}+Season_{j}+Parity_{k}+e_{ijkl}$$
where *Y_ijkl_* is the observed value of lactation traits; *μ* is the population mean; *Month_i_* is the calving age effect; *Season_j_* is the calving season effect; *Parity_k_* is the parity effect; and *e_ijkl_* is the residual.

### 2.5. Estimation of Individual Breeding Values

The single-trait animal model of the DMUAI module of the derivative-free approach to multivariate analysis (DMU, Version 6) software was used to estimate the variance components of lactation traits for all individuals in the test population, and the variance-covariances of the obtained fixed effects were used as a priori values to calculate the estimated breeding values of the individuals by using the DMU4 module. The models for estimating the variance components of lactation traits were as follows:Y=Xz+Wu+e
where *Y* is the observed value, *z* is the fixed effect (calving age, calving season, and parity), *X* is the correlation matrix of fixed effects, *u* is the random effect (individual ID), *W* is the correlation matrix of random effects, and *e* is the residual.

### 2.6. Population Structure and Kinship Identification

Principal component analysis (PCA) was performed using Tassel 5.2.8 software to examine the existence of stratification in the test population. Based on the first three principal components of the PCA results, the population structure map was drawn using the ggplot package of R 4.1 software. On the basis of the population structure map, we observed whether the population was stratified and added the results of the first three principal components to the correlation analysis of the mixed linear model. The kinship matrix was constructed using Tassel 5.2.8 software with genotype data and visualized using R 4.1 software, and a kinship heat map was drawn to observe the existence of kinship between individuals. The kinship matrix results were added to the mixed linear model for association analysis.

### 2.7. Genome-Wide Association Study

In this study, individual estimated breeding values were used as phenotypes with Tassel 5.2.8 software and correlated with SNP gene chip typing data for analysis. The mixed linear model used in this experiment was as follows:Y=Sα+Qν+Zu+e
where *Y* is the individual breeding value of lactation traits; α is the fixed effect of SNP markers; *v* is the population structure effect; *u* is the residual polygenic effect, *u*~N (0.$2K\sigma_{a}^{2}$), where *K* is the kinship matrix and $\sigma_{a}^{2}$ is the genetic variance; *e* is the residual; and *S*, *Q*, and *Z* are the association matrices of *α*, *v*, and *u*, respectively. The results obtained from the association analysis were corrected for multiple hypothesis testing using the Bonferroni method. Briefly, N (81,203) was the total number of filtered SNPs, and the genomic level significance (6.1574 × 10^−7^, 0.05/N) and chromosome level significance (1.2315 × 10^−5^, 1/N) were used as thresholds.

### 2.8. Identification of Candidate Genes and Functional Enrichment Analysis

In this study, Ensembl UMD3.1 (http://asia.ensembl.org/Bos_taurus/Info/Index/, accessed on 2 October 2021) database was used to locate genes within 1 Mb upstream and downstream of significant SNP sites. To better understand the biological processes, gene ontology (GO) terms and encyclopedia of genes and genomes (KEGG) pathways were annotated, visualized, and integrated for discovery on the basis of relevant genes by using the DAVID 6.8 database (https://david.ncifcrf.gov/, accessed on 18 October 2021). A *p*-value < 0.05 was the threshold for significant enrichment of GO terms and KEGG pathways.

## 3. Results

### 3.1. Descriptive Statistics and Estimated Breeding Values of Lactation Ttraits

Table 1 summarizes the descriptive statistics for the three lactation traits, MY, FP, and PP, of the test population with means of 9807.07, 3.54, and 3.19, respectively, and coefficients of variation (CV) of 15.64%, 13.56%, and 7.21%, respectively. Table 2 shows the results of the ANOVA for lactation trait fixed effects. From the results, it is clear that the calving age effect had a highly significant effect on MY and FP (*p* < 0.001) and no significant effect on PP. The parity effect had no significant effect on MY and PP and a significant effect on FP (*p* < 0.01). The calving season effect had a highly significant effect on both MY (*p* < 0.001) and FP (*p* < 0.01) and no significant effect on PP. Therefore, correction for calving age and calving season effects was required to estimate MY individual breeding values, and correction for calving age, parity, and calving season effects was required to estimate FP individual breeding values, while no correction for fixed effects was performed for PP individual breeding values. The statistical results of the estimated breeding values for the lactation traits MY, FP, and PP are shown in Table 3, with means of 70.25, −0.03, and −0.01 and estimated heritability of 0.07, 0.12, and 0.20, respectively.

### 3.2. Population Stratification

The PCA results (Figure 1A) showed that the test population was divided into several subpopulations of different sizes with significant stratification. Therefore, the results of PCA were included as covariates in the association analysis model and population stratification based on the results of PCA analysis was considered in the mixed linear model. The kinship mapping is shown in Figure 1B. Many individuals in the test population were related to each other, and there were eight larger families consisting of many related individuals. Therefore, the kinship of individuals within the test population needed to be considered in the association analysis, and the effect of kinship between individuals on the association analysis results was corrected by including the kinship matrix as a random effect in the mixed linear model of association analysis.

### 3.3. Genome-Wide Association Results

The Manhattan plot and quantile–quantile plot (Q-Q plot) of the association analysis are shown in Figure 2. According to the Manhattan plot, numerous SNP loci significantly associated with the FP trait were located on BTA14. These significant loci form peaks of SNP loci, as observed from the results. Combined with Table 4, it can be inferred that these significant loci were mainly situated within a 0.5 Mb window ranging from 0.4 to 0.9 Mb on BTA14, with a total of 15 significant SNP loci, which was the smallest range and the most concentrated distribution of SNP loci in this association analysis. Based on the principle of linkage disequilibrium, it is possible that some of these significant SNP loci were false positives, showing strong linkage disequilibrium with the true associated SNP loci, leading to significant results. Therefore, it is necessary to comprehensively discuss these loci considering their physical positions, associated genes, and previous studies. Regarding the SNP loci significantly associated with the PP trait, they primarily resided in a 2 Mb window on BTA11 (3 SNPs). On the other hand, the three significant SNP loci on BTA21 were relatively scattered. No significant SNP loci associated with the MY trait were identified in this study. Observing the Manhattan plot and Q-Q plot of the MY trait, it can be observed that the Q-Q plot of the MY trait exhibited a good fit. This may be attributed to the conservative correction imposed by the Bonferroni method in multiple hypothesis testing, which could filter out some SNP loci that might have significant associations with the MY trait, particularly for large-scale chip data. Another possibility is that the lack of association between the MY trait and significant SNP loci could be due to the relatively small sample size in the association analysis.

Table 4 results demonstrate that a total of 23 SNP loci were found to be significantly associated with lactation traits in this study, with 13 SNP loci showing significance at the genome level (*p* < 6.1574 × 10^−7^) and 10 SNP loci exhibiting significance at the chromosome level (*p* < 1.2315 × 10^−5^). Among the SNP loci associated with FP trait, there were a total of 16 loci, with 15 of them located on BTA14. The most significantly associated SNP with the FP trait was identified as ARS-BFGL-NGS-4939, which is situated within the diacylglycerol O-acyltransferase 1 gene (*DGAT1*). Within the *PLEC* gene, two SNP loci were found to be significantly associated with the FP trait. For the PP trait, a total of 7 SNP loci were significantly associated, including 3 loci on BTA11, all located within the *SNRNP200* gene, and 3 loci on BTA21, respectively, within the *CERS3*, *AP3S2*, and *GABRA5* genes. The most significantly associated SNP with the MP trait was identified as Hapmap32084-BTA-147824, which is situated within the *CERS3* gene.

### 3.4. GO and KEGG Analyses for Lactation Traits

We performed GO and KEGG pathway enrichment analysis on a total of 145 genes located within a 1 Mb range upstream and downstream of the significant SNP loci. GO analysis revealed a significant enrichment (*p* < 0.05) for a total of 45 GO terms, including 16 biological process terms, 4 cellular component terms, and 25 molecular function terms (Figure 3, Appendix A and Table 5). The KEGG pathways analysis revealed that three pathways were enriched. Among them, two pathways (bta01100: Metabolic pathways and bta00480: glutathione metabolism) were significantly enriched, whereas the third pathway (bta01230: biosynthesis of amino acids) did not have a significant *p*-value (*p* = 0.067; Table 4). *IDH2* was enriched in three pathways; *ANPEP*, *OPLAH*, *GPT*, and *PYCR3* were enriched in two pathways.

## 4. Discussion

High-temperature and -humidity environments can contribute to metabolic imbalances and cause heat stress in cows, who experience a range of physiological and behavioral responses under heat stress conditions [8]. Heat stress decreases dry matter intake, affects nutrient metabolic activity, and reduces MY, FP, and PP in cows [4,6]. Heat stress not only affects milk production and quality, it also affects the overall health of the cow, affecting normal physiology, metabolism, hormones, and the immune system [13]. In this study, GWAS analysis was performed for lactation traits in Holstein cows under high temperature and high humidity conditions in Southern China. The GWAS results obtained 23 SNP loci that were significantly associated with lactation traits. Of the significantly associated SNPs, 16 were associated with FP and 7 were associated with PP. No SNPs significantly associated with MY traits were obtained, which indicates the limitations of this study, such as the sample size of the test population and the restricted genotyping platform. Among the significantly associated SNPs, 11 loci were previously reported to be significantly associated with lactation traits. For example, a study found that Chr14_1765835, Chr14_1757935, Chr14_2022745, and Chr14_1653693 were significantly associated with MY and PP [14]. Other studies found that BovineHD1400000216, BovineHD1400000275, Bo-vineHD1400000206, BovineHD1400000204, and BovineHD1400000271 were significantly associated with MY, FP, and PP in dairy cows [15,16]. Among the 16 SNP sites significantly associated with FP, 14 SNP sites were located within a window of approximately 0.5 Mb between 0.4 and 0.9 Mb on BTA14, and 68 genes were associated within 1 Mb upstream and downstream of this window. Numerous studies have shown the presence of a large number of quantitative trait loci (QTLs) and variant sites that affect lactation traits in dairy cows in the range of 0.4 to 0.9 Mb on BTA14. A SNP locus (rs137205809) which significantly affects FP was found at 0.7 Mb of BTA14 [17]. A large number of SNP sites significantly related to FP were found at 0.7 Mb of BTA14 [18]. Pausch et al. [19] performed targeted sequence padding of four candidate causal variant locus regions and association analysis of milk components and successfully identified SNP sites at 0.6 Mb on BTA14 that were significantly associated with FP. These findings are very close to the results of the present study, so QTLs that significantly affect FP in Chinese Holstein cows can be hypothesized to exist within a window of approximately 0.5 Mb between 0.4 and 0.9 Mb on BTA14.

In this study, novel SNP loci associated with lactation traits were discovered and have not been previously reported. Among these loci, the most significant one was Chr14_1699016 (*p* = 8.36 × 10^−8^), which is located 384 bp away from *VPS28* on BTA14 and approximately 37 kb away from the previously reported BovineHD1400000216 locus. Some studies have reported that *VPS28* gene was highly expressed in breast tissue and significantly associated with FP [10,20]. The SNP loci BovineHD1400000282 and BovineHD1400000287 are located within the PLEC gene on chromosome 14, which has been reported to play a crucial role in lactation processes [21]. Furthermore, studies have suggested that the PLEC gene may be associated with disease resistance or alleviation of heat stress [22]. DB-813-seq-rs110906821 is located on BTA14 at 1844 bp away from the *EXOSC4* gene, which has been reported to be significantly associated with colostrum and serum protein concentrations in Chinese Holstein cows [23]. BovineHD1200022127 is located within the *HS6ST3* gene on BTA12, a member of the gene family involved in heparin metabolism. Heparin decreases the rate of lipoprotein lipase degradation in adipocytes and promotes adipocyte differentiation [24]. ARS-BFGL-NGS-4300 is located within the *AP3S2* gene on BTA21. A previous study found that *AP3S2* expression was significantly upregulated in Sahiwal cattle at 4, 24, and 48 h after heat stress, suggesting that the AP3S2 gene may be associated with heat stress [25]. BovineHD2100000735 is located within the *GABRA5* gene on BTA21. GABRA5 is a subunit of gamma-aminobutyric acid A receptor, which plays a crucial role in maintaining ion balance. Ion balance is vital for lactation in cows [26]. In conclusion, the three newly discovered heat stress-related SNP loci (BovineHD1400000282, BovineHD1400000287, and ARS-BFGL-NGS-4300) identified in this study are potentially associated with lactation traits in cows under heat stress conditions.

In this study, the biological processes, cellular components and molecular functions obtained from GO analysis did not directly affect lactation traits. KEGG analysis identified 16 genes (*DGAT1*, *CYC1*, *CERS3*, *GPAA1*, *IDH2*, *GPT*, *PYCR3*, *OPLAH*, *HDDC3*, *ALDH1A3*, *MAN2A2*, *NAPRT*, *CYP11B1*, *ANPEP*, *GPAT2*, and *GFUS*) that were enriched in three pathways, namely metabolic pathways, glutathione metabolism, and biosynthesis of amino acids. Metabolic pathways are crucial for lactation in cows [15], while glutathione plays a significant role in antioxidant stress response and nutrient metabolism [27]. Moreover, amino acid biosynthesis serves as an important source of protein in milk [28]. Numerous studies have reported the association between the *DGAT1* gene and milk fat synthesis [29,30]. Furthermore, the *DGAT1* gene is a major gene in inflammatory response and lipid metabolism, encoding an enzyme that catalyzes the last step of triacylglycerol synthesis [31], and significantly affecting energy metabolism during lactation in dairy cows [32]. The *CYC1* gene encodes a subunit of the cytochrome bc1 complex, which is associated with redox processes, and it has been reported that the *CYC1* gene is an important candidate for albumin synthesis [23]. Albumin is known to have anti-inflammatory and antioxidant functions [33], so the *CYC1* gene may be associated with anti-inflammatory responses. The *IDH2* gene encodes isocitrate dehydrogenase. In the absence of ATP-citrate lyase, ruminants can use isocitrate dehydrogenase to form NADPH, which raises energy for cells, and the excess energy is converted to fatty acids, which are then stored as fat [34]. GWAS analysis and pathway analysis of fat deposition traits in Nello cattle revealed the relationship between *IDH2* gene and fat deposition [35]. The *GPT* gene encodes glutamic-pyruvate transaminase. Habeeb et al. [36] conducted a study comparing the physiological and biochemical changes in Egyptian native cows and introduced cows under hot summer conditions, finding that serum levels of glutamate aminotransferase were significantly higher in native cows than in introduced cows under hot conditions, suggesting that the *GPT* gene may be associated with heat stress in dairy cows. The *PYCR3* gene encodes a protein belonging to the pyrroline-5-carboxylic acid reductase family, which has been shown to be involved in genotoxic, inflammatory, and oxidative stress responses [37,38]. The *OPLAH* gene encodes an enzyme involved in the γ-glutamyl cycle and is responsible for converting the degradation product of glutathione, 5-oxoproline, back to glutamate [39]. Since 5-oxoproline is an oxidative stress inducer [40], *OPLAH* functions as an anti-oxidative stress agent by removing this metabolite. Many studies have identified the *OPLAH* gene as a candidate gene for milk yield traits in Holstein cows [14,41]. The *ALDH1A3* gene encodes the third enzyme of the aldehyde dehydrogenase family 1, which is involved in the conversion of retinaldehyde to retinoic acid and has detoxifying and antioxidant functions [42]. The *NAPRT* gene encodes nicotinic acid phosphate ribosyltransferase, which has a key role in the acute inflammatory response [43]. The *CYP11B1* gene encodes steroid 11-β-hydroxylase, which is involved in steroid hormone biosynthesis. Many studies have shown that the *CYP11B1* gene is associated with milk yield and milk fat percentage in dairy cows [44]. The *GFUS* gene encodes GDP-L-amylose synthase, which is involved in inflammatory and immune responses by activating the immune network [45], and it has also been shown that *GFUS* is associated with milk protein percentage and milk albumin content in dairy cows [23,46]. Heat stress affects the anti-inflammatory response and immune function of cows by reducing the production of antibodies, cytokines, chemokines, and heat shock proteins [47]. In addition, cows exposed to heat stress have shown reduced MY and FP as well as PP [6]. In this study, GWAS analysis of lactation traits in Chinese Holstein cows under high temperature and humidity in Southern China was performed to screen for genes reported to be directly related to lactation traits (*DGAT1*, *IDH2*, *CYP11B1*, and *GFUS*) and candidate genes not reported to be directly related to lactation traits but associated with antioxidant, anti-stress, inflammatory, immune response, and detoxification processes (*CYC1*, *GPT*, *PYCR3*, *OPLAH*, *ALDH1A3*, *NAPRT*, and *GFUS*).

## 5. Conclusions

In summary, this study screened 23 SNPs that were significantly associated with lactation traits in Chinese Holstein cows in the high temperature and humidity region of Southern China. Among them, three SNPs were associated with heat stress. In addition, this study identified QTLs that significantly affected milk fat percentage in Chinese Holstein cows within a window of approximately 0.5 Mb between 0.4 and 0.9 Mb on BTA14. Ten most promising candidate genes (*DGAT1*, *IDH2*, *CYP11B1*, *GFUS*, *CYC1*, *GPT*, *PYCR3*, *OPLAH*, *ALDH1A3*, and *NAPRT*) were revealed based on KEGG analysis. The results of this study are useful for molecular marker-assisted breeding of lactation traits in Chinese Holstein cows under high temperature and humidity regions to accelerate genetic improvement.

## Figures and Tables

**Figure 1 animals-13-02545-f001:**
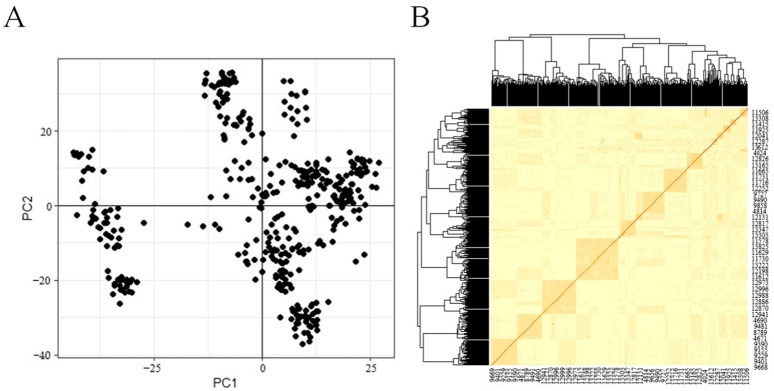
(**A**) The PCA plot of top two principal components. (**B**) The kinship matrix diagram.

**Figure 2 animals-13-02545-f002:**
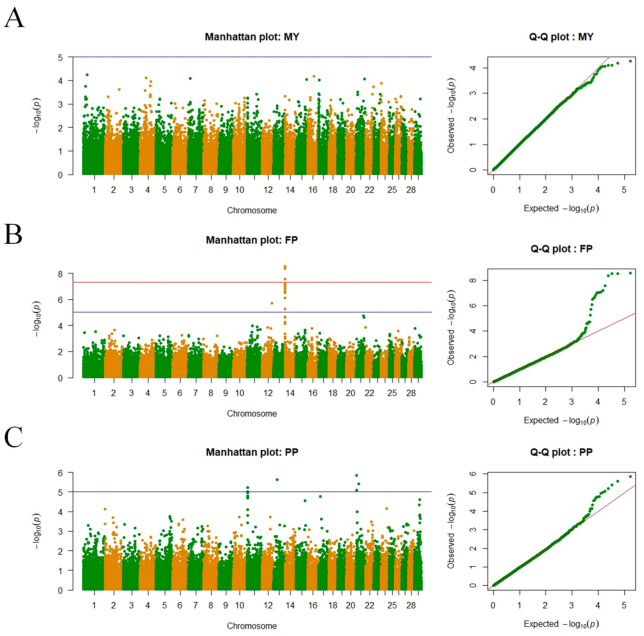
Manhattan plots and Q–Q plots from GWAS for MY, FP, and PP traits in Chinese Holstein cows. (**A**) MY: milk yield; (**B**) FP: milk fat percentage; (**C**) PP: milk protein percentage. In the Manhattan plot, above the blue horizontal line indicates a significant SNP chromosome level and above the red horizontal line indicates a significant SNP genomic level.

**Figure 3 animals-13-02545-f003:**
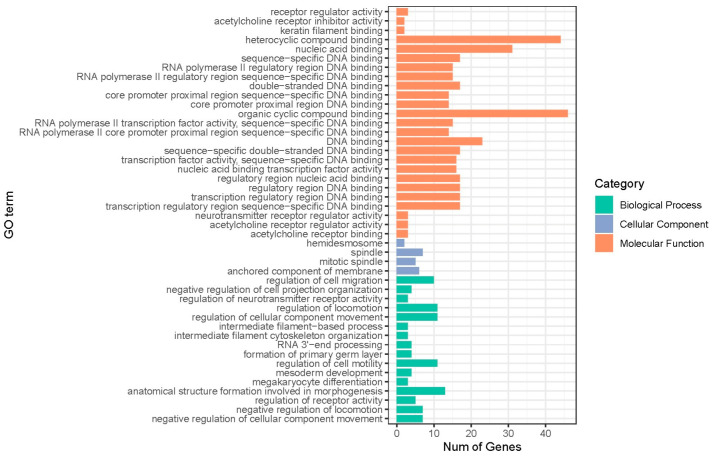
GO analysis of regional candidate genes for lactation traits.

**Table 1 animals-13-02545-t001:** Descriptive statistics of lactation traits ^1^.

Traits	*n*	Mean	Min	Max	SD	CV (%)
MY	392	9807.07	4870.42	14205.58	1538.60	15.64
FP	392	3.54	1.97	5.07	0.48	13.56
PP	392	3.19	2.34	3.76	0.23	7.21

^1^ MY, milk yield; FP, milk fat percentage; PP, milk protein percentage. CV, coefficient of variation.

**Table 2 animals-13-02545-t002:** Analysis of variance components of lactation traits phenotypes ^1^.

Item	MY	FP	PP
Calving age	<0.001 ***	<0.001 ***	0.797
Parity	0.101	0.002 **	0.296
Calving season	<0.001 ***	0.007 **	0.054

^1^ MY, milk yield; FP, milk fat percentage; PP, milk protein percentage. ** indicates significant at the *p* < 0.01 level, and *** indicates significant at the *p* < 0.001 level.

**Table 3 animals-13-02545-t003:** Statistics of the estimated breeding values for lactation traits ^1^.

Traits	*n*	Mean	Min	Max	SD	Heritability
MY	392	70.25	−1013.32	1118.44	408.94	0.07
FP	392	−0.03	−0.45	0.41	0.16	0.12
PP	392	−0.01	−0.34	0.26	0.10	0.20

^1^ MY, milk yield; FP, milk fat percentage; PP, milk protein percentage.

**Table 4 animals-13-02545-t004:** Genome-wide significant SNPs associated with FP and PP traits and nearest candidate genes.

Traits	SNP Name	BTA	Position (bp)	*p*-Value	Nearest Gene Name	Distance (bp)
FP	ARS-BFGL-NGS-4939	14	609,870	2.85 × 10^−9^	*DGAT1*	Within
	Chr14_1765835	14	580,019	2.96 × 10^−9^	*SLC52A2*	Within
	BovineHD1400000216	14	550,784	3.12 × 10^−9^	*CPSF1*	Within
	Chr14_1757935	14	572,120	2.74 × 10^−8^	*ADCK5*	1622
	Chr14_2022745	14	831,004	6.44 × 10^−8^	*GRINA*	938
	Chr14_1699016	14	513,203	8.36 × 10^−8^	*VPS28*	384
	BovineHD1400000282	14	859,251	9.51 × 10^−8^	*PLEC*	Within
	BovineHD1400000287	14	883,732	9.51 × 10^−8^	*PLEC*	Within
	BovineHD1400000275	14	2,019,390	1.00 × 10^−7^	*TSNARE1*	Within
	ARS-BFGL-NGS-57820	14	465,742	1.46 × 10^−7^	*FOXH1*	3390
	BovineHD1400000206	14	494,621	2.25 × 10^−7^	*TONSL*	1650
	Chr14_1653693	14	468,124	2.52 × 10^−7^	*FOXH1*	1008
	BovineHD1400000204	14	487,527	3.05 × 10^−7^	*CYHR1*	Within
	DB-813-seq-rs110906821	14	758,854	8.04 × 10^−7^	*EXOSC4*	1844
	BovineHD1200022127	12	73,935,024	2.02 × 10^−6^	*HS6ST3*	Within
	BovineHD1400000271	14	810,116	5.73 × 10^−6^	*SPATC1*	3725
PP	Hapmap32084-BTA-147824	21	6,095,419	1.40 × 10^−6^	*CERS3*	Within
	Hapmap25132-BTA-96391	13	27,232,535	2.37 × 10^−6^	*FZD8*	445,190
	ARS-BFGL-NGS-4300	21	21,278,489	3.94 × 10^−6^	*AP3S2*	Within
	ARS-BFGL-NGS-40264	11	2,382,261	6.04 × 10^−6^	*SNRNP200*	Within
	BovineHD2100000735	21	4,285,575	8.19 × 10^−6^	*GABRA5*	Within
	BovineHD1100000821	11	2,385,285	9.53 × 10^−6^	*SNRNP200*	Within
	BovineHD1100000817	11	2,381,644	1.10 × 10^−5^	*SNRNP200*	Within

**Table 5 animals-13-02545-t005:** KEGG analysis of regional candidate genes for lactation traits.

Term	Description	Gene Count	%	*p*-Value	Gene
bta01100	Metabolic pathways	16	11.1	0.03333	*CERS3, DGAT1, GPAA1, IDH2, GPT, PYCR3, OPLAH, HDDC3, ALDH1A3, MAN2A2, NAPRT, CYP11B1, ANPEP, CYC1, GPAT2, GFUS*
bta00480	Glutathione metabolism	3	2.1	0.04989	*ANPEP, IDH2, OPLAH*
bta01230	Biosynthesis of amino acids	3	2.1	0.06663	*IDH2, GPT, PYCR3*

## Data Availability

The variation data reported in this paper have been deposited in the genome variation map (GVM) in National Genomics Data Center, Beijing Institute of Genomics, Chinese Academy of Sciences and China National Center for Bioinformation, under accession number GVM000516 (https://bigd.big.ac.cn/gvm/getProjectDetail?Project=GVM000516, accessed on 8 April 2023).

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
