# Peer review of "Genome-Wide Association Study of Lactation Traits in Chinese Holstein Cows in Southern China"

_animals, 2023, doi:10.3390/ani13152545_

Round 1
Reviewer 1 Report
There are a lot of GWAS studies on dairy cattle. The 392 Chinese Holstein cows seem small in this kind of study. What are this work's unique discoveries and contributions to dairy cattle breeding?
There are a lot of GWAS studies on dairy cattle. The 392 Chinese Holstein cows seem small in this kind of study. What are this work's unique discoveries and contributions to dairy cattle breeding?
Reviewer 2 Report
The document presents GWAS for economically important traits, methods were applied correctly and results were discussed according to Ensemble UMD3.1 annotations. I suggest also searching annotations on ARS-UCD1.2.
This work's novel trait (heat stress) should be highlighted in the introduction, results, and conclusions.
Reviewer 3 Report
In this study, the whole genome of 392 Holstein cows in southern China was analyzed for lactation traits. 23 single nucleotide polymorphic loci significantly associated with lactation traits and 10 key genes associated with lactation fat percentage, milk yield, antioxidant activity, stress resistance, inflammation and immune response were screened. The study is very relevant to the development of an effective selection and breeding program for Chinese Holstein cows in high temperature and humidity areas, but the study has some issues that need further confirmation.
I have only a few minor suggestions for the authors to consider:
1. All cows for the same dairy farm does not represent the southern region of China, and should be sampled and tested in multiple regions.
2. The sample size should include the different stages of lactation, such as the beginning, peak and end stages of lactation, and the different times of milking.
3. line 234: "Figure 2" to "Figure 3".
4. The materials and methods mentioned calving age, calving frequency, and calving season, but the results did not analyze the differences in lactation traits by calving age, calving frequency, and calving season.
5. Further validation of the differential genes in KEGG analysis.
6. The Chinese description appears several times in the text, such as in lines 44, 50, 52, 248, 250, 257, 259, 261, 310, and 313 where someone finds it.
Reviewer 4 Report
The main objective of the manuscript entitled “Genome-wide Association Study of Lactation Traits in Chinese Holstein Cows in Southern China” is QTL mapping of lactation traits in Chinese Holstein Cows. There are some major concerns regarding statistical methods and I think that most of the analyses must be repeated in a more efficient way.
Here, I attached my major concerns:
1. Lines 65-66: “All animals were housed under the same conditions, and they were free of disease, healthy, and in good mental condition.” I wonder how the authors evaluated that all animals were in good mental condition?! If authors applied a scientific approach to evaluating mental condition of animals, then it must be explained. Also, it must be mentioned what diseases were tested. How authors diagnosed that all animals were healthy? These must be mentioned in Material and Methods section.
2. Lines 101-111: number of individuals must be mentioned. How many phenotyped individuals were considered for estimation of breeding values? Pedigree depth and number of individuals in pedigree must be mentioned.
3. Table 3: I wonder how many individuals were used for variance component estimation? Estimated heritability for MY is very low.
4. Authors used EBVs of individuals as dependent variable in GWAS model. Using raw EBVs in GWAS models can be problematic due to accounting parent averages in estimating breeding values. Therefore, EBVs should be adjusted for their parent averages through deregressing of EBVs. I highly suggest using deregressed EBVs instead of raw EBVs. Please consider if the number of individuals is low in BLUP, then using adjusted phenotypes might be more efficient than EBVs.
Round 2
Reviewer 4 Report
The authors applied some of my suggestions. However, there are some critical concerns regarding statistical analyses in this study. The population size of animals is very low to have a reliable variance component estimation and consequently for having reliable EBVs. Therefore, they must use adjusted phenotypes for GWAS analyses as former studies applied adjusted phenotypes for GWAS:
https://doi.org/10.3390/ani11071927
-
https://doi.org/10.1186/s12863-019-0725-0
- https://www.frontiersin.org/articles/10.3389/fgene.2021.799664/full
Also, the authors did not mention the depth of pedigree in manuscript.
Round 3
Reviewer 4 Report
This manuscript is publishable in its present form.